# Loss of Tid1/DNAJA3 Co-Chaperone Promotes Progression and Recurrence of Hepatocellular Carcinoma after Surgical Resection: A Novel Model to Stratify Risk of Recurrence

**DOI:** 10.3390/cancers13010138

**Published:** 2021-01-04

**Authors:** Kuan-Yang Chen, Yi-Hsiang Huang, Wan-Huai Teo, Ching-Wen Chang, Yu-Syuan Chen, Yi-Chen Yeh, Chieh-Ju Lee, Jeng-Fan Lo

**Affiliations:** 1Institute of Clinical Medicine, National Yang-Ming University, Taipei 11221, Taiwan; DAA13@tpech.gov.tw; 2Institute of Neuroscience, National Chengchi University, Taipei 11605, Taiwan; 3Department of Gastroenterology, Ren-Ai Branch, Taipei City Hospital, Taipei 10629, Taiwan; 4Division of Gastroenterology and Hepatology, Department of Medicine, Taipei Veterans General Hospital, Taipei 11217, Taiwan; ssbugi@gmail.com; 5Institute of Oral Biology, National Yang-Ming University, Taipei 11221, Taiwan; whteo@gm.ym.edu.tw (W.-H.T.); flower750904@gmail.com (C.-W.C.); joy12132@gmail.com (Y.-S.C.); 6Laboratory of Human Carcinogenesis, National Cancer Institute, Bethesda, MD 20892, USA; 7Department of Pathology and Laboratory Medicine, Taipei Veterans General Hospital, Taipei 11217, Taiwan; ycyeh2@vghtpe.gov.tw; 8Department of Dentistry, Taipei Veterans General Hospital, Taipei 11217, Taiwan; 9Cancer Progression Research Center, National Yang-Ming University, Taipei 11221, Taiwan

**Keywords:** hepatocellular carcinoma, Tid1, Nrf2, HBV, biomarker and recurrence

## Abstract

**Simple Summary:**

Tid1 acts as a tumor suppressor in various cancer types, however, its role in hepatocellular carcinoma (HCC) remains unclear. Here, we observed a low protein level of Tid1 in poorly differentiated HCC cell lines. The expression of Tid1 affected the malignancy in human HCC cell lines; meanwhile the protein level of Nrf2 was negatively regulated by Tid1. In multivariate analysis, using immunohistochemical (IHC) assay in 210 HCC cases, we found the tumor size > 5 cm, multiple tumors, presence of vascular invasion, low Tid1 expression in the non-tumor part, and high Nrf2 expression in the non-tumor part, were independently associated with worse recurrence-free survival (RFS). A scoring system by integrating the five clinical and pathological factors predicts the RFS among HCC patients after surgical resection. In summary, Tid1 plays a prognostic role for surgically resected HCC.

**Abstract:**

Tid1, a mitochondrial co-chaperone protein, acts as a tumor suppressor in various cancer types. However, the role of Tid1 in hepatocellular carcinoma (HCC) remains unclear. First, we found that a low endogenous Tid1 protein level was observed in poorly differentiated HCC cell lines. Further, upregulation/downregulation of Tid1 abrogated/promoted the malignancy of human HCC cell lines, respectively. Interestingly, Tid1 negatively modulated the protein level of Nrf2. Tissue assays from 210 surgically resected HCC patients were examined by immunohistochemistry (IHC) analyses. The protein levels of Tid1 in the normal and tumor part of liver tissues were correlated with the clinical outcome of the 210 HCC cases. In multivariate analysis, we discovered that tumor size > 5 cm, multiple tumors, presence of vascular invasion, low Tid1 expression in the non-tumor part, and high Nrf2 expression in the non-tumor part were significant factors associated with worse recurrence-free survival (RFS). A scoring system by integrating the five clinical and pathological factors predicts the RFS among HCC patients after surgical resection. Together, Tid1, serving as a tumor suppressor, has a prognostic role for surgically resected HCC to predict RFS.

## 1. Introduction

Hepatocellular carcinoma (HCC) comprises over 80% of cases of primary liver cancer and is the fourth leading cause of cancer-related death worldwide, and, especially, the second in terms of death for men [1,2]. Chronic hepatitis virus infection such as hepatitis B virus (HBV) and hepatitis C virus (HCV) are the most common etiological risk factors for HCC [3,4]. Surgical resection, radiofrequency ablation (RFA), and liver transplantation are curative treatments for HCC. However, the recurrence rate is high even after curative resection or RFA, and liver transplantation is limited by shortage of organ source [5,6]. Although antiviral treatment for HBV and HCV has been reported to be able to reduce recurrence, the risk of recurrence is still unpredictable [7,8,9]. Hence, identification of a potential biomarker is essential for predicting the recurrence of HCC after curative treatment.

Tumorous imaginal disc 1 (Tid1) is the mammalian homolog to the Drosophila tumor suppressor protein Tid56 [10]. The Tid1 gene, including 12 exons, encodes two alternative splice forms, Tid1-L and Tid1-S, which are responsible in encoding two cytosolic (hTid50 and hTid48) and the other two mitochondrial (hTid43 and hTid40) proteins, respectively. Tid1-L fully incorporates all exons and Tid1-S decoded from an in-frame deletion of 50 amino acids that correspond precisely to exon 5 [11,12]. Tid1, also called DnaJ homolog subfamily A member 3 (DNAJA3), belongs to the heat shock protein (Hsp) 40 family and serves as a co-chaperone and regulatory factor for the heat shock protein 70 (Hsp70) to sustain embryonic-cell survival [13,14], T cell development [15], muscular development [16], and apoptosis [17]. Tid1, acting as a tumor suppressor, interacts with EGFR/HSP70/HSP90 by DnaJ domain and causes EGFR degradation in non-small cell lung cancer (NSCLC) [18] and in head and neck squamous cell carcinoma (HNSCC). Overexpression of Tid1 in HNSCC cell lines enables the inhibition of in vitro malignancy and in vivo xenotransplantation tumorigenicity and metastasis [19]. Additionally, we reported that the HNSCC patients with higher expression of Tid1 have better overall survival [19,20]. Most recently, we also discovered that Tid1 can function as a tumor suppressor in gastric cancer progression [21]. It has been reported that Tid1 plays an important role to maintain mitochondrial DNA [21], membrane potential [22], and cristae structure [23], and to modulate the intracellular reactive oxygen species (ROS) [24]. The Warburg effect is a signature of carcinogenesis, in that mitochondrial dysfunction plus higher intracellular reactive oxygen species (ROS) are associated with it [25]. Thus, loss of Tid1 accompanied by abnormal ROS may contribute to tumor progression in HCC. Investigation of the impact of Tid1 in HCC may reveal a new strategy and new biomarker for treatment.

Nuclear erythroid 2-related factor 2 (Nrf2), is a CNC-bZIP cytosolic transcription factor that can act against oxidative stress [26]. Oxidative stress is also a key etiological risk factor for HCC [27]. The transient activation of Nrf2 offers protection during hepatocytes that are exposed to chemical carcinogens [28], but in contrast, persistent activation of Nrf2 can drive the oncogenic process [29,30]. In HCC, sustained Nrf2 activation leads to cellular proliferation and resistance against drugs [31,32]. We recently show that ROS-independent ER stress can mediate the Nrf2 activation to promote cancer stemness [33]. These studies provide evidence that Nrf2 as an oncogene in HCC may be used as a potential prognosis marker.

It is reasonable that Tid1, a tumor suppressor, regulates the mitochondrial function on ROS homeostasis. During tumorigenesis, downregulation of Tid1 can cause abnormal ROS, which can be counteracted by the antioxidant transcriptional factor, Nrf2. In this line of research, we aimed to investigate the role of Tid1 and Nrf2 in the outcomes of HCC after surgery by in vitro study and samples from HCC patients.

## 2. Results

### 2.1. Tid1 Functions as a Tumor Suppressor in HCC Cell Lines In Vitro

We first analyzed the expression level of Tid1 protein from datasets of the Oncomine database platform derived from 19 normal control and 38 HCC cases. As expected, that Tid1 may act as a tumor suppressor in HCC, we discovered that the Tid1 protein level was significantly reduced in HCC compared to normal liver tissues (Figure 1A). In order to evaluate the tumor suppressor role of Tid1 in HCC, we further examined the endogenous Tid1 protein expression in human HCC cell lines. Using immunoblot analyses, we found that a low level of endogenous Tid1 protein was observed in poorly differentiated HCC cells such as Mahlavu and SK hep1 with the exception of Hep3B, which is a well-differentiated cell line (Figure 1B). It has been reported that Hep3B cell may harbor the HBV in which the HBX protein has been reported to regulate the protein level of Tid1 [30]. Inversely, we found that high expression of Tid1 was observed in moderated or well-differentiated HCC cells (Huh7 and HepG2) (Figure 1B).

Tid1 maintains and regulates the normal mitochondrial function and ROS homeostasis [21,24]. During tumorigenesis, downregulation of Tid1 can cause abnormal ROS, which can be counteracted by the antioxidant transcriptional factor, Nrf2. Interestingly, we observed that the protein level of endogenous Nrf2 was negatively associated with the protein level of Tid1, except in the Hep3B cells (Figure 1B). To further characterize the anticancerous role of Tid1 in HCC, we overexpressed human hTid1-L and -S isoform in SK hep1 cells, respectively. As expected, SK hep1 cells with Tid1 overexpression showed the downregulation of Nrf2 protein (Figure 1C). In opposite, when we depleted the endogenous Tid1 protein by small hairpin RNA interference (shRNAi) in HepG2 cells, we observed the upregulation of Nrf2 protein (Figure 1D). Additionally, the colony-forming ability of the Tid1 overexpressing cells was repressed (Figure 1E), and of the Tid1 depleted cells was enhanced (Figure 1F), respectively. Together, these results indicated that Tid1 regulated the in vitro malignancy of human HCC cells and played an inhibitory role on Nrf2. 

### 2.2. Expression Correlation between the Tid1 and Nrf2 Transcripts

Next, to verify the correlation of mRNA expression between DNAJA3 (encodes Tid1) and NFE2L2 (encodes Nrf2) genes, we downloaded the published liver cancer cohort from the Oncomine database and performed Expression Correlation analysis. As shown in Figure 2A, surprisingly, expression of DNAJA3 and NFE2L2 mRNA was positively correlated in both normal and cancer tissues in cohort GSE15420, and the correlation coefficient R reached 0.48 with a *p* value of 1.7 × 10^−13^ in normal tissues, and in cancer tissues, the R reached to 0.22 with a *p* value of 0.0015. However, there was no significant correlation between the comparison of cancer tissue to normal tissue or normal tissue to cancer tissue. By analysis of the liver cancer cohort GSE17967, a negative correlation between Tid1 and Nrf2 in HCC cancer tissue but not in cirrhosis tissues (Figure 2B) was observed. Together, these inconsistent results shown in Figure 1A and Figure 2A,B suggest a reciprocal regulation between Tid1 and Nrf2, mediated mainly in translation or post-translation rather than in transcription in HCC.

### 2.3. Clinicopathological Features and Protein Level of Tid1 and Nrf2 of HBV-HCC and HCV-HCC

The clinicopathological characteristics of the 210 HCC subjects with liver tissue samples from the Taiwan Liver Cancer Network are summarized in Table 1. The mean age of diagnosis was 59 years, 74% of the subjects were male, and 41% had cirrhosis, with 105 cases of HBV infection and 105 cases of HCV infection, while 2 cases were dual HBV and HCV infection. Mean of tumor size was 6 cm, and 33% of subjects had multinodular tumors. Most tissues were diagnosed with moderately differentiation (70%).

### 2.4. Tid1 Expression in Non-Tumor and Tumor Part of TNM-T Stage Specific HCC

To determine the correlation between tumor stage and the protein levels of Tid1 in HCC patients, we performed IHC staining on sections of the 210 HCC liver tissues. Tid1 protein was significantly downregulated in tumor tissues, compared to the non-tumor part (Table 2). Further, high expression of Tid1 in non-tumor tissues was significantly associated with T stage (*p* = 0.002), especially, in HBV-associated HCC (*p* = 0.017) and marginal association in HCV-associated HCC (*p* = 0.093) (Table 3). The data indicated that as the tumor progressed, the expression of tumor suppression gene, Tid1, in the non-tumor part was enhanced. While Tid1 expression in tumor part had a trend to correlate with tumor stage, and reduction of Tid1 expression was significantly correlated with tumor T stage in HCV-associated HCC (*p* = 0.019) (Table 4). Consistent with the findings of Figure 1B, the Tid1 expression profile had substantial correlation with HCC malignancy.

### 2.5. Tid1 Expression in Non-Tumor Liver Tissues of Cirrhotic and Non-Cirrhotic State of HCC

We further compared the expression of Tid1 protein between patients with or without cirrhosis in non-tumorous liver tissues. The protein level of Tid1 in cirrhotic samples was significantly lower than that in non-cirrhotic samples, both in HBV-HCC and HCV-HCC (Table 5), as cirrhosis is an important risk factor of HCC. This finding implies that Tid1 expression is also highly associated with HCC development.

### 2.6. Tid1 as a Prognostic Factor for Recurrence-Free Survival in HCC

To further examine whether Tid1 could serve as a prognostic biomarker to predict the recurrence of HCC after surgical resection, distinct univariate and multivariate analyses were conducted (Figure 3 and Table 6). As analyzed, patients with high AFP (≥400 ng/mL, hazard ratio (HR):1.556, *p* = 0.019), multiple tumor number (tumor number >1, HR:2.111, *p* < 0.001), presence of microvascular vascular invasion (HR:1.798, *p* = 0.001), advanced T stage (T2 or T3, HR: 2.085 and 2.582, *p* = 0.010 and *p* < 0.001, respectively), advanced staging (stage III and IV versus stage I and II, HR:2.574, *p* < 0.001), low expression level of Tid1 in the non-tumorous part (*p* = 0.054), and high Nrf2 expression in the non-tumorous part (*p* = 0.050) had an unfavorable recurrence-free survival (RFS) in univariate analysis. In multivariate analysis, tumor size, tumor number, presence of microvascular vascular invasion, Nrf2 expressions in the non-tumorous part (HR=1.527, 95% CI = 1.023–2.280, *p* = 0.038), and Tid1 expression in the non-tumorous part (HR = 0.500, 95% CI = 0.330-0.756, *p* = 0.001) were independent factors associated with recurrence.

As shown in Figure 4A, using a prognostic score by incorporation of the five factors established, the risk of HCC recurrence could be divided into four groups according to the stratification, with median recurrence-free survival of >120 months for low risk (score 0–1), 29.4 months for median risk (score 2), 13.5 months for high risk (score 3), and 6.2 months for very high risk (score 4–5) patients (*p* < 0.0001) (Figure 4B).

## 3. Discussion

Previous studies suggest that Tid1 is a tumor suppressor in various cancer types, but this role had not been previously evaluated in HCC [18,19,20]. In this study, we performed IHC to confirm that HCC tissue had a lower expression level of Tid1 than that in normal liver; overexpression of Tid1 could inhibit colony formation, and downregulation of Tid1 would promote in vitro malignancy in HCC cell lines. However, Nrf2, as an oncogene, its mRNA level was not well inversely associated with Tid1 mRNA, as confirmed by the Oncomine platform. In addition, both overexpression and depletion of Tid1 protein abrogated the protein level of Nrf2 in HCC cells. Nevertheless, both Tid1 and Nrf2 protein levels in the non-tumorous part of liver were significant factors associated with RFS in the 210 HCC patients after surgical resection. Moreover, a novel scoring system by incorporation of clinical factors and Tid1, and Nrf2 expression could stratify patients into four categories of risk in recurrence of HCC after surgical resection.

Interaction of Tid1 and viral oncoproteins reveals the role of Tid1 in virus-induced carcinogenesis [34,35,36]. Functional J-domain of viral oncoproteins is important for viral replication and cellular transformation. The effect of interaction of DnaJ protein Tid1 and viral oncoproteins is controversial. Tid1 interacts with HPV E7 and Epstein–Barr virus encoded BARF1 accelerates maturation and secretion of these proteins, as well as maintains biological functions [34,37]. Degradation of HBV core protein and HBx via ubiquitination by binding with Tid1 represses HBV replication [36]. In addition, several Tid1-regulated proteins were found involved in hepatocarcinogenesis, such as EGFR [18,20], ErbB2 [38], Myc [39], Ras [40], VEGF, and Hif1α [41]. Previously, we reported that HNSCC patients with low expression of Tid1 demonstrated a poor prognosis. Furthermore, we demonstrate that by overexpressing Tid1 in HNSCC cells inhibit the in vitro cell proliferation, migration, invasion, and anchorage-independent growth. Additionally, ectopic overexpression of Tid1 is shown to suppress in vivo xenotransplantation tumorigenicity and diminish Galectin-7-mediated metastasis [19,20]. Abovementioned that Tid1 may play as a tumor suppressor in HCC. In this study, significant reduction of Tid1 is observed in tumor tissues compared to the non-tumor part (Figure 1A). Patients with high Tid1 in the non-tumor part of the liver have a favorable recurrence-free survival in HCC (Figure 3D). These results show Tid1 expression in the non-tumor part of liver can serve as a novel prognostic factor to predict the patient’s prognosis after surgery. Our data also found that as the tumor progressed, the expression of Tid1 in the non-tumor part was mostly enhanced (Table 2 and Table 3). This association with expression of Tid1 in the non-tumor part might reflect a protective anti-tumor mechanism of humans in face of tumor progression.

Tumor suppression functions of Tid1 rely on degradation of ErbB2 in breast cancer [42], and attenuation of EGFR/Akt/Erk signaling in HNSCC and in NSCLC [18,20]. Mitochondria translocation of p53 by Tid1 triggers intrinsic mitochondria-mediated cell death in p53 wild-type MCF-7 and HCT-116 cells [43,44]. In hepatitis virus-related HCC, most patients harbored mutated p53 [45]; this may explain why Tid1 lost its suppression function in the tumor part and could be a tumor suppressor against cancer spreading in the normal part of the liver. In this study, we also demonstrate Tid1 suppresses anchorage-independent growth in p53-wild-type HCC cells (Figure 1C).

Nrf2 is found as a key transcription regulator for antioxidant and detoxification [46]. Activation of Nrf2 is observed in liver cells, such as hepatic stellate cells and Kupffer cells, as well as in parenchymal hepatocytes [47,48]. It was reported that Nrf2 played a protective role in hepatic inflammation, fibrosis, and hepatocarcinogenesis, through its target gene induction [49]. However, accumulative evidences indicate that Nrf2 is abundantly expressed in cancer cells, including HCC. It was reported that Nrf2 activation is related to proliferation, invasion, and chemoresistance in HCC [31,50,51]. Our results revealed that there were significant correlations between the expressions of Nrf2 with recurrence of HCC (Table 6 and Figure 3E). This suggests that Nrf2 may play an oncogenic role in HCC. Surprisingly, we also observed a negative correlation between Tid1 and Nrf2. Expression of Nrf2 was affected in Tid1 overexpressed-SK hep1 and Tid1 depleted-HepG2 cells, respectively (Figure 1C,D). Nevertheless, we analyzed the mRNA expression of two liver cancer cohorts from the open-source database and discovered that the correlation of Tid1 and Nrf2 was inconsistent. Of note, the regulation between Tid1 and Nrf2 proteins could be in the post-translation level but not the transcription level, and future exploration to study how Tid1 negatively regulates the Nrf2 protein is necessary.

Surgical resection is the better therapy for HCC but with a limitation of high probability of recurrence [52]. Tumor size, tumor number, and microvascular invasion are well-known factors associated with recurrence after resection of HCC. [53] Interestingly, the recurrent patterns categorized by Tid1 and Nrf2 in the non-tumor part indicate that the tumor suppressor effect of Tid1 might play a more important role for late recurrence, while the oxidative stress effect of Nrf2 is responsible for both early and late recurrence (Figure 3D,E). Development new biomarkers or combination with clinical and pathological features may be a bright way for prognosis determination. Molecular alteration in non-tumor part tissues could provide more information for clinical application [54]. Expression of low Tid1 and high Nrf2 in the non-tumor part showed significance with worse recurrence-free survival. Combination with clinical and molecular features showed a valuable contribution to stratify the risk of recurrence after surgical resection of HCC. In summary, Tid1, serving as a tumor suppressor, has a prognostic role to predict RFS for surgically resected HCC.

## 4. Materials and Methods

### 4.1. Cell Lines

Human hepatocyte carcinoma cell lines, Mahlavu, Huh7, Hep3B, HepG2, and SK hep1, were purchased from ATCC and maintained with complete medium containing 10% fetal bovine serum (FBS) for further experiments.

### 4.2. Human Specimens

The tissue arrays of 210 surgically resected HCC specimens were provided from the Taiwan Liver Cancer Network (TLCN No. 150104) with available clinical characteristics and outcomes [55]. The tumors and non-tumor tissues of liver were used for Tid1 and Nrf2 expression analysis by immunohistochemistry (IHC) staining to analyze the association between Tid1, Nrf2 protein, and clinical parameters, as well as prognosis of HCC.

### 4.3. Data Collection of DNAJA3 (Tid1) and NFEL2 (Nrf2) mRNA Expression from Open Source Database

Expression of Tid 1 differential transcriptomes in normal and HCC tissues was obtained from the Oncomine public data portal (www.oncomine.org). The datasets GSE15420 and GSE17967 were randomly selected. The correlation between DNAJA3 (Tid1) and NFEL2 (Nrf2) mRNA expression was also compared from Oncomine. Normalized RNA-Seq data (log2) were used as gene expression value and for measuring the correlation coefficient.

### 4.4. Western Blotting

Western blotting was used to measure Tid1 protein level (clone RS13, Thermo Scientific, Fremont, CA, USA) in HCC cell lines; GAPDH protein level served as loading control. The procedures of Western blotting were described previously [18].

### 4.5. Overexpression by hTid1-L or -S Isoform

The plasmids with pRK5 backbone containing C-terminal tag of HA and harboring the full-length cDNAs of Tid1 long form (Tid1L) or short form (Tid1S), respectively, were constructed for ectopic expression, and have been described [13,19,20].

### 4.6. Downregulation by Small Hairpin RNA Interference (shRNAi)

The lentiviral shRNA plasmids for the knockdown of Tid1 protein level were purchased from National RNAi Core Facility (Academia Sinica, Taipei, Taiwan), as follows: ShDNAJA3-1: Clone ID: TRCN0000008775 and ShDNAJA3-2: Clone ID: TRCN0000273884. ShLuc vector (Clone ID: TRCN0000072266) against luciferase was used as a negative control. The method of lentivirus production and cell infection was described in the manufacturer’s protocol.

### 4.7. Cell Anchorage-Independent Growth Assay

Anchorage-independent growth assay was performed as described previously [19]. Briefly, each well of a 6-well culture dish was coated with 1 mL bottom agar mixture (DMEM, 10% (vol/vol) FBS, 0.6% (wt/vol) agar, 1% (vol/vol) penicillin–streptomycin) [31,50,51]. After the bottom layer was solidified, 1 × 10^4^ plasmid-transfected (pRK5-Ctrl, pRK5-hTid1-L-wt and pRK5-hTid1-S-wt) [13] or virus-infected (shLuc, shDNAJA3-1 and shDNAJA3-2) cells were plated, subsequently followed by topping approximately 1 mL of top agar-medium mixture (DMEM, 10% (vol/vol) FBS, 0.3% (wt/vol) agar, 1% (vol/vol) penicillin–streptomycin) to each well. The dishes were cultured at 37 °C with 5% CO_2_ for 2 weeks. Cells grown on the plates were stained with 0.005% crystal violet for 1 h, and then the cell colonies were quantitated over five fields per well for 15 fields in triplicate experiments.

### 4.8. Immunohistochemistry (IHC)

IHC was performed as described previously [19,20,42]. The primary antibody against Tid1 protein (clone RS13, Thermo Scientific, Fremont, CA, USA) was performed according to a previous report [40]. The H scores of Tid1 were defined by a single pathologist (Figure 5). In brief, H-score method is a semiquantitative approach defined as the sum of the products obtained by multiplying staining intensity (0, 1, 2, and 3) by the percentage of positively stained cells with respect to each staining intensity. H-scores ranged from 0 to 300. Intensity was defined as follows: 0 for no detectable staining, 1+ for weak reactivity mainly detectable at high magnification (20–40× objective), and 2+ or 3+ for intense (moderate or strong, respectively) reactivity easily detectable at low magnification (4× objective). Ref. [56] Tid1 expression of the non-tumor part was subgrouped as low (≤75) and high (>75). Tid1 expression of the tumor part was subgrouped as low (≤47.25) and high (>47.25). The primary antibody against Nrf2 protein (NCBI NP_001138884.11; NM_001145412.21, Biorbyt LLC, San Francisco, CA, USA) was performed according to manufacturer instruction.

### 4.9. Statistical Analysis

The association between Tid1 and Nrf2 was compared by the chi-square test. Continuous variables were compared by the Mann–Whitney U test. Fisher’s exact test and Pearson chi-square analysis were applied to assess categorical data. The Kaplan–Meier method with log-rank test was used for recurrence-free survival (RFS) analyses. Analysis of prognostic factors for RFS was performed using the Cox proportional hazards model. Variables that achieved statistical significance (*p* < 0.05) or those close to significance (*p* < 0.1) by univariate analysis were subsequently included in the multivariate analysis. For all analyses, *p* < 0.05 was considered statistically significant. All statistical analyses were performed using IBM SPSS Statistics (SPSS 27 for Windows) and Excel (Microsoft Office Professional Plus 2013).

## 5. Conclusions

Tid1 plays a novel prognostic role in HCC after surgery. Additionally, suppression of tumorigenesis and cancer progression by Tid1 may imply that this co-chaperone protein could be a promising prognostic marker and potential therapeutic target for HCC. Combination of clinical features (tumor size and number, and vascular invasion) with Tid1 and Nrf2 expression in the non-tumor part of livers could provide a novel and useful strategy to stratify the extremely high-risk group of recurrence after surgical resection.

## Figures and Tables

**Figure 1 cancers-13-00138-f001:**
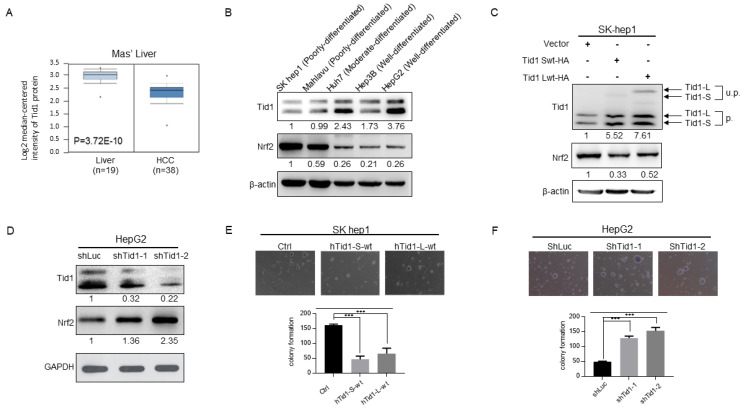
Endogenous Tid1 protein level in normal liver, hepatocellular carcinoma (HCC) tissues, HCC cell lines; and ectopic overexpression or downregulation of Tid1 mediating the malignancy of HCC cells in vitro. (**A**) Comparison of Tid1 protein level in normal livers and HCC was analyzed from Oncomine database. The selection criteria are descripted in Material and Methods section. (**B**) Expression of endogenous Tid1 protein of five HCC cell lines (Mahlavu, Huh7, Hep3B, HepG2, and SK hep1) were examined by immunoblot assay using an antibody against Tid1. (**C**) The immunoblot analysis showed that overexpression of Tid1 (Tid1-L-wt-HA or Tid1-S-wt-HA) abrogated the Nrf2 protein in the transfected SK hep1 cells (u.p. unprocessed, p. processed). (**D**) The immunoblot analysis showed the depletion of Tid1 by shRNAi in both HepG2 cells; the Nrf2 protein level was analyzed. The signal of β-actin or GAPDH was used as loading control. The colony formation ability of the (**E**) SK hep1 or (**F**) HepG2 cells under distinct condition was collected, as shown in the representative images. The bar graphs show the amount of colonies. The data are the mean ± SD from three independent experiments and analyzed by Student’s *t*-test (***, *p* < 0.001). Detailed information about the western blotting can be found at Appendix A

**Figure 2 cancers-13-00138-f002:**
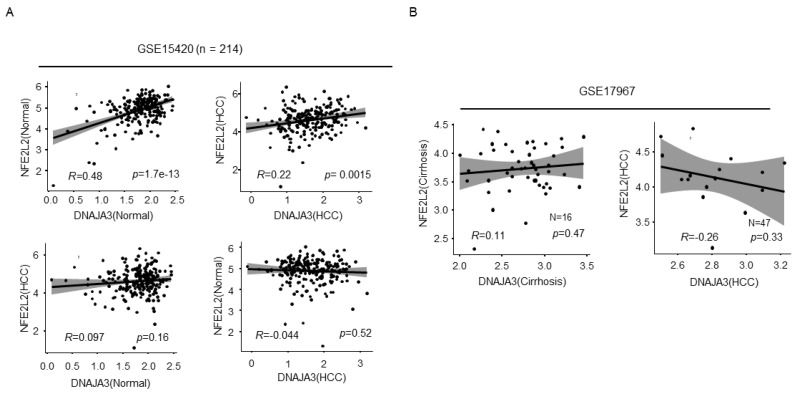
Expression correlation between the NDAJA3 (Tid1) and NFE2L2 (Nrf2) transcripts in normal liver, cirrhosis, and HCC. Expression correlation analysis between DNAJA3 and NFE2L2 genes in liver cancer cohort (GSE15420 (**A**) and GSE17967 (**B**)). HCC indicates liver cancer tissue, Normal indicates non-cancer tissue, and Cirrhosis indicates tissue with cirrhosis but no liver cancer. *R* indicates correlation coefficient, *p* indicates significance, and *n* indicates sample capacity.

**Figure 3 cancers-13-00138-f003:**
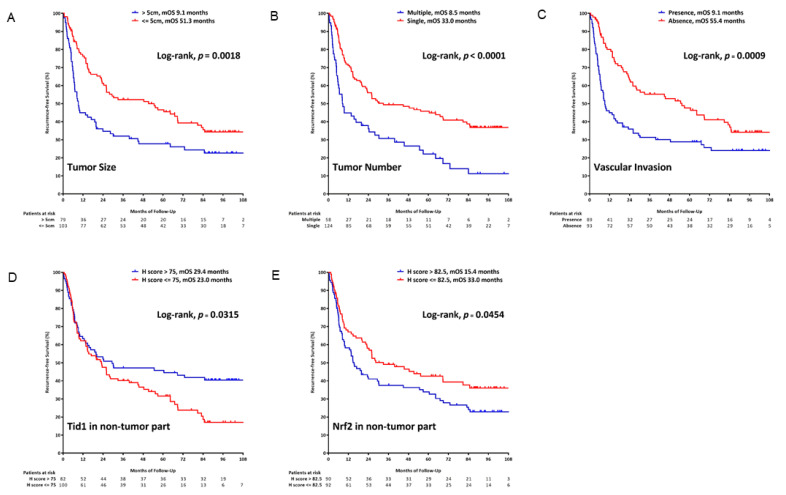
Recurrence-free survival (RFS) of the HCC patients categorized by (**A**) tumor size of 5 cm; (**B**) single or multiple tumors; (**C**) presence of microvascular invasion or not; (**D**) Tid 1 expression level in the non-tumor part; and (**E**) Nrf2 expression level in the non-tumor part. The Kaplan–Meier method with log-rank test was used for RFS analyses.

**Figure 4 cancers-13-00138-f004:**
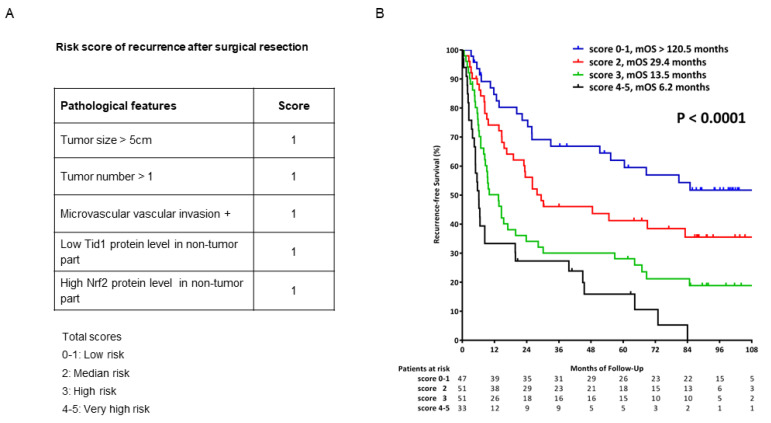
Recurrence-free survival of the HCC patients stratified into four groups based on the scoring system. (**A**) Prognostic score established by incorporation of the five factors: tumor size, tumor number, vascular invasion, low Tid1 protein level, and high Nrf2 protein level in the non-tumor part. (**B**) Recurrence-free survival of HCC patients according to the established prognostic score.

**Figure 5 cancers-13-00138-f005:**
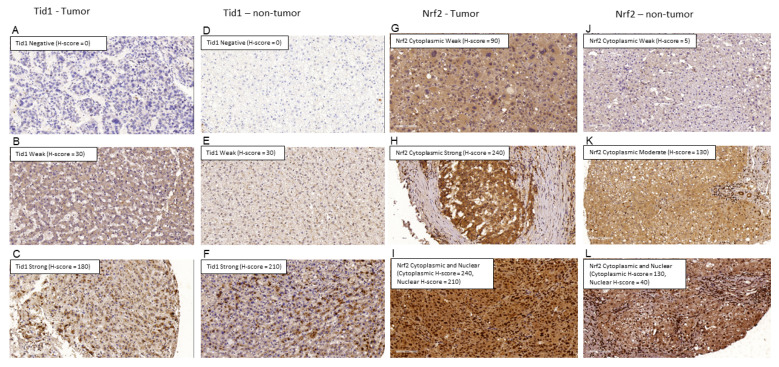
Examples of Tid1 expression in the tumor and non-tumor part. (**A**) Negative Tid1 expression in the tumor part. (**B**) Weak Tid1 expression in the tumor part. (**C**) Strong Tid 1 expression in the tumor part. (**D**) Negative Tid1 expression in the non-tumor part. (**E**) Weak Tid1 expression in the non-tumor part. (**F**) Strong Tid1 expression in the non-tumor part. (**G**) Weak Nrf2 expression in tumor part. (**H**) Strong Nrf2 expression in tumor part. (**I**) Cytoplasmic and nuclear Nrf2 expression in tumor part. (**J**) Weak Nrf2 expression in non-tumor part. (**K**) Moderate Nrf2 expression in non-tumor part. (**L**) Cytoplasmic and nuclear Nrf2 expression in non-tumor part.

**Table 1 cancers-13-00138-t001:** Clinical characteristics of the 210 patients with hepatocellular carcinoma (HCC).

Variables	HCC Patients (*n* = 210)
Age, years, mean ± SD	59.23 ± 13.12
Male sex, *n* (%)	156 (74.3)
Drinking, *n* (%)	67 (31.9)
Dual HBV and HCV infection, *n* (%)	2 (1.0)
Alpha-fetoprotein, ng/mL, mean ± SD	14,833.18 ± 64,194.02
Tumor size, cm, mean ± SD	6.15 ± 4.10
Tumor number, *n* (%)	
*n* = 1/*n* > 1/Diffuse or infiltrative	141/68/1 (67.1/32.4/0.5)
Cirrhosis, *n* (%)	86 (41.0)
Differentiation, *n* (%)	
Well/Moderately/Poorly	14/147/49 (6.7/70.0/23.3)
BCLC stage, *n* (%)	
A/B/C-D	73/17/120 (34.8/8.1/57.1)
TNM stage, *n* (%)	
T1/T2/T3	73/24/113 (34.8/11.4/53.8)

HCC, hepatocellular carcinoma; HBV, hepatitis B virus; HCV, hepatitis C virus; SD, standard deviation; BCLC, Barcelona Clinic Liver Cancer.

**Table 2 cancers-13-00138-t002:** TID1 expression (H score) in tumor and non-tumor tissues of paired HCC patients divided by TNM-T stage.

T Stage	T1 + T2			T3		
H Score	Tumor	Non-Tumor	*p* Value	Tumor	Non-Tumor	*p* Value
HCC (all) *	10 (0–165)	55 (0–285)	<0.001	20 (0–185)	85 (0–255)	<0.001
HBV-HCC **	0 (0–165)	57.5 (0–285)	<0.001	0 (0–185)	97.5 (0–255)	<0.001
HCV-HCC ***	30 (0–130)	55 (0–185)	0.028	55 (0–145)	70 (13–160)	0.004

H score presented as median (range); * missing paired samples *n* = 9; ** missing paired samples *n* = 1; *** missing paired samples *n* = 8.

**Table 3 cancers-13-00138-t003:** Tid1 expression in non-tumor part of liver tissues.

TNM-T Stage/HCC Differentiation	Low H Score ≤ 75, *n* (%)	High H Score > 75, *n* (%)	*p* Value
HCC, *n* = 210			
TNM-T stage			0.002
T1 and T2, *n* = 93	63 (56.3)	30 (33.3)	
T3, *n* = 109	49 (43.8)	60 (66.7)	
Differentiation			0.395
Well/Moderate, *n* = 157	90 (80.4)	67 (74.4)	
Poorly, *n* = 45	22 (19.6)	23 (25.6)	
HBV-HCC, *n* = 105			
TNM-T stage			0.017
T1 and T2, *n* = 42	26 (53.1)	16 (29.1)	
T3, *n* = 62	23 (46.9)	39 (70.9)	
Differentiation			0.426
Well/Moderate, *n* = 87	43 (87.8)	44 (80.0)	
Poorly, *n* = 17	6 (12.2)	11 (20.0)	
HCV-HCC, *n* = 105			
TNM-T stage			0.093
T1 and T2, *n* = 51	37 (58.7)	14 (40.0)	
T3, *n* = 47	26 (41.3)	21 (60.0)	
Differentiation			0.361
Well/Moderate, *n* = 70	47 (74.6)	23 (65.7)	
Poorly, *n* = 28	16 (25.4)	12 (34.3)	

**Table 4 cancers-13-00138-t004:** Tid1 expression in liver tumors.

TNM-T Stage/HCC Differentiation	Low H Score ≤ 47.25	High H Score > 47.25	*p* Value
HCC, *n* = 210			
TNM-T stage			0.096
T1 and T2, *n* = 95	72 (50.0)	23 (37.1)	
T3, *n* = 111	72 (50.0)	39 (62.9)	
Differentiation			0.109
Good/Moderate, *n* = 158	115 (79.9)	43 (69.4)	
Poorly, *n* = 48	29 (20.1)	19 (30.6)	
HBV-HCC, *n* = 105			
TNM-T stage			0.197
T1 and T2, *n* = 42	40 (42.6)	2 (20.0)	
T3, *n* = 62	54 (57.4)	8 (80.0)	
Differentiation			0.361
Good/Moderate, *n* = 87	77 (81.9)	10 (100.0)	
Poorly, *n* = 17	17 (18.1)	0	
HCV-HCC, *n* = 105			
TNM-T stage			0.019
T1 and T2, *n* = 53	32 (64.0)	21 (40.4)	
T3, *n* = 49	18 (36.0)	31 (59.6)	
Differentiation			0.200
Good/Moderate, *n* = 71	38 (76.0)	33 (63.5)	
Poorly, *n* = 31	12 (24.0)	19 (36.5)	

**Table 5 cancers-13-00138-t005:** Tid1 expression in non-tumor liver tissue of cirrhotic and non-cirrhotic status of HCC.

H Score, Median (Range)	Cirrhosis	Non-Cirrhosis	*p* Value
HCC *	60 (0–210)	80 (0–285)	<0.001
HBV-HCC **	60 (0–210) (*n* = 41)	85 (0–285) (*n* = 63)	0.016
HCV-HCC ***	58 (0–108) (*n* = 43)	70 (5–185) (*n* = 55)	0.006

H score presented as median (range); * missing samples, Cirrhosis/Non-Cirrhosis, *n* = 2/6; ** missing samples, Cirrhosis/Non-Cirrhosis, *n* = 0/1; *** missing samples, Cirrhosis/Non-Cirrhosis, *n* = 2/5.

**Table 6 cancers-13-00138-t006:** Univariate and multivariate analyses of factors associated with recurrence-free survival in 210 HCC patients.

Characteristics		Univariate			Multivariate I			Multivariate II		
		HR	95% CI	*P*	HR	95% CI	*P*	HR	95% CI	*P*
Age (years)	>50 vs. ≤50	1.080	0.726–1.607	0.703			NA			NA
Sex	Male vs. Female	0.902	0.601–1.353	0.617			NA			NA
Alpha-fetoprotein (ng/mL)	≥400 vs. <400	1.556	1.074–2.255	0.019			NS			NS
Liver Cirrhosis	Presence vs. Absence	1.285	0.905–1.825	0.161			NS			NS
Underlying Hepatitis	HCV vs. HBV	1.128	0.795–1.600	0.499			NA			NA
Tumor size (cm)	>5 vs. ≤5	1.573	1.109–2.230	0.011	1.618	1.074–2.439	0.021			NA
Tumor number	>1 vs. ≤1	2.111	1.474–3.024	<0.001	1.606	1.071–2.408	0.022			NA
Vascular invasion	Presence vs. Absence	1.798	1.268–2.552	0.001	1.658	1.104–2.490	0.015			NA
TNM-T stage	T1	1.000	-				NA	1.000	-	
	T2	2.085	1.196–3.635	0.010						NS
	T3	2.582	1.700–3.922	<0.001				2.508	1.593–3.949	<0.001
	TREND			<0.001						<0.001
Pathology stage	3–4 vs. 1–2	2.574	1.788–3.704	<0.001			NA			NA
Differentiation	Good	1.000	-				NA			NA
	Moderate	1.549	0.717–3.348	0.265						
	Poor	1.539	0.677–3.499	0.303						
	TREND			0.534						
IHC expression, H score										
Tid1, non-tumor part	>75 vs. ≤75	0.699	0.486–1.007	0.054	0.500	0.330–0.756	0.001	0.510	0.337–0.771	0.001
Tid1, tumor part	>47.25 vs. ≤47.25	1.397	0.960–2.033	0.081			NS			NS
Nrf2-nucleus, non-tumor part	>5 vs. ≤5	1.082	0.663–1.763	0.753			NA			NA
Nrf2-nucleus, tumor part	>40 vs. ≤40	1.244	0.868–1.783	0.235			NA			NA
Nrf2-cytoplasma, non-tumor part	>82.5 vs. ≤82.5	1.421	0.999–2.021	0.050	1.527	1.023–2.280	0.038	1.531	1.027–2.282	0.036
Nrf2-cytoplasma, tumor part	>95 vs. ≤95	1.490	1.049–2.116	0.026			NS			NS

## Data Availability

Publicly available datasets (Oncomine) were analyzed in this study. This data can be found here: [www.oncomine.org]. The other data presented in this study are available on request from the corresponding author. The data are not publicly available due to their massive file size.

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
