# Peer review of "Loss of Tid1/DNAJA3 Co-Chaperone Promotes Progression and Recurrence of Hepatocellular Carcinoma after Surgical Resection: A Novel Model to Stratify Risk of Recurrence"

_cancers, 2021, doi:10.3390/cancers13010138_

Round 1

Reviewer 1 Report

The paper appear definitively improved and now it's worty to be published

Reviewer 2 Report

There are no problem in revise answer comments. 

This manuscript is a resubmission of an earlier submission. The following is a list of the peer review reports and author responses from that submission.

Round 1

Reviewer 1 Report

1) Isn't the expression of TiDd1 in none HCC tissue different depending on the state of cirrhosis?

2)  Why is TiD1 expression level associated with the T stage? It is need to discussion.

3) What is H score? What is the definition? It reqire to explain into methotology. 

 How to definition of low H score and High H score?

3) In multivariate analysis, what meaning  of  multivariate I and II,  in table 5.

4) Also table 5, Why dose authors  select at  age 50yrs, AFP 400 ng/ml and tumor size 5cm etc.

Were those numbers calculated by cut-off values?  

5) SPSS version is too old.

6) Dose author have data of after chemotherapy ? 

Reviewer 2 Report

In this manuscript, Chen and colleagues studied in tumoral and peritumoral samples of hepatocellular carcinoma (HCC) the s the clinicopathological significance of the expression of the oncosupressor Tid1 and of the transcription factor Nrf2. For these studies, the Authors used different approaches, among which immunohistochemistry (IHC) on human CC samples, cell biology, using HCC cell lines, and analysis of clinical data.

The most important results reached by the Authors are that A) in poor differentiated HCC cells, Tid1 is downregulated, and that, B) the manipulation of Tid1, both silencing or upregulation, induce the deregulation of tumor cell proliferation; C) there is a negative correlation between Tid1 and Nrf2 expression, in vitro. D) in peritumoral human samples, the expression of low levels of Tid1 and high levels of Nrf2 is an indication of poor disease free survival.

The paper is potentially of interest, in particular concerning the analysis of human samples and the clinical part. Unfortunately, the in vitro section is very poor and presents several flows dampen significantly the enthusiasm for this paper.

Major concerns

  1. Is not clear why the Authors decided to analyze the expression of Nrf2 and its correlation with Tid1, among several other proteins putatively involved: which is the rationale at the basis of this choice? Please discuss thoroughly.
  2. Authors claim that Tid1 induced Nrf2 modulation, but is not explained or hypothesized a mechanism, the simple correlation of the two data could be only incidental.
  3. Authors claimed that Tid1 modulation drives HCC recurrence but in vitro data do not support evidences obtained on human patients database: Authors showed that Tid1 downregulation is an indicator of worse outcome when expressed at low level on non-tumoral areas is expected the presence of normal hepatocytes, but they used for in vitro data, neoplastic cell lines.
  4. In line with point 3, to claim that low Tid1 and high Nrf2 expression are fundamental to induce an aggressive tumor behavior in neoplastic cells, how the Authors explain that seems there is no significance (apart for Nrf2-cytoplasmic expression in tumor part) between their expression in and recurrence-free survival (table 5)? And what about overall survival of the patients?
  5. Authors generate a potentially interesting novel method to stratify patients (figure 3): this could be reported in the title of the publications.
  6. 1: the Authors have to report the plots showing the effectiveness of the down-regulation of overexpression of Tid1 following genetic manipulation.

Minor concerns

  1. The “results” section is confusedly written and difficult to follow.
  2. in “graphical abstract”, the legend of the Kaplan-Meier curve is very small and difficult to read.
  3. Figure 2: the reciprocal expression of Tid1 and Nrf2 could be showed also in HepG2 cells.
  4. Table 3: please explain the choice of the value selected to discriminate high and low subgroups.